# Effect of Intramuscular Tramadol on the Duration of Clinically Relevant Sciatic Nerve Blockade in Patients Undergoing Calcaneal Fracture Fixation: A Randomized Controlled Trial

**DOI:** 10.3390/healthcare11040498

**Published:** 2023-02-08

**Authors:** Marek Janiak, Grzegorz Gorniewski, Rafal Kowalczyk, Piotr Wasilewski, Piotr Nowakowski, Janusz Trzebicki

**Affiliations:** 11st Department of Anesthesiology and Intensive Care, Medical University of Warsaw, 02-005 Warsaw, Poland; 2Department of Anesthesiology and Intensive Care Education, Medical University of Warsaw, 02-007 Warsaw, Poland; 3Department of Orthopedics and Traumatology, Medical University of Warsaw, 02-005 Warsaw, Poland; 4Department of Anesthesiology and Intensive Care, Gruca Orthopedic and Trauma Teaching Hospital, 05-400 Otwock, Poland

**Keywords:** tramadol, sciatic nerve block, calcaneal fracture fixation

## Abstract

Background: Calcaneal fracture fixation can generate severe postoperative pain and analgesia can be supported by a sciatic nerve block. However, following resolution of the sensory blockade, rebound pain may ensue. The aim of this study was to assess whether an incidental finding of two patients with an extension of the sciatic nerve block beyond 24 h following 100 mg of intramuscular tramadol administration could be confirmed. Methods: Thirty-seven patients scheduled for a calcaneal intramedullary fixation (Calcanail^®^) were randomly divided into two groups. The tramadol group (*n* = 19) received a sciatic nerve block with 20 mL of 0.25% bupivacaine and a concomitant dose of 100 mg of intramuscular tramadol, while the control group (*n* = 18) received an identical sciatic nerve block with concomitant injection of normal saline (placebo). All patients had a spinal anesthesia with light sedation for the procedure. The time to first analgesic request defined as appearance of any pain (NRS > 0) was assessed as the primary endpoint with a clinically relevant expected result of at least 50% elongation in sensory blockade. Results: The median time to first analgesic request from time of blockade in the tramadol group was 670 min compared with 578 min in the control group. The result was clinically not relevant and statistically not significant (*p* = 0.17). No statistical difference could be demonstrated in the time to first opioid request, although a trend for opioid sparing in the tramadol group could be seen. Total morphine consumption in the first 24 h was also statistically insignificant (the tramadol group 0.066 mg kg^−1^ compared with 0.125 mg kg^−1^ in the control group). In conclusion, intramuscular tramadol does not extend the duration of analgesia of a sciatic nerve block following a calcaneal fracture fixation beyond 2 h and an opioid sparing effect could not be demonstrated in this trial.

## 1. Introduction

Calcaneal fractures may account for only 1–2% of all fractures, but about 71% are intraarticular and most require surgical reduction and fixation for better functional and anatomical recovery [1,2]. However, postoperative pain in the first 24 h following open calcaneal fracture surgery has been assessed as the most severe among 179 surgical procedures with a median numerical rating score (NRS) of 6.8 [3]. Hence, it may be prudent to supplement basic anesthesia with a regional block covering the innervation of the hindfoot such as a single-shot sciatic nerve block. The duration of such a block may not ensure pain control for the first 24 h after the procedure and severe rebound pain may ensue. The use of several adjuvants has been proven to extend the duration of the sensory blockade. Dexamethasone has proven efficacy as an adjuvant to nerve blocks when administered both perineurally and intravenously with at least 1.5 times when compared with nerve block with only local anesthetic [4,5]. Tramadol has been studied as an adjuvant in perineural and systemic co-administration with a nerve block, with inconsistent results to date. No trial assessing the use of intramuscular tramadol in conjunction with a sciatic nerve block is known to the authors of this study.

Based on an incidental finding in our institution of an effective sciatic nerve blockade with over 24 h in two patients following calcaneal fracture repair that received intramuscular tramadol prior to surgery, we designed a randomized trial with the aim to investigate whether tramadol can prolong the sensory blockade by at least 1.5 times for clinical relevance providing the ‘overnight’ analgesic benefit.

## 2. Materials and Methods

The prospective single-center, randomized trial was conducted at the 1st Department of Anesthesiology and Intensive Care, Clinical University Centre Hospital, Warsaw, Poland. This trial was approved by the ethics review board of the Warsaw Medical University (Approval number KB/128/2017, Chairperson Prof. Zbigniew Wierzbicki) and registered with data safety authorities with study registry under ClinicalTrials.gov, NCT03477851. The study protocol complied with the principles laid down in the Declaration of Helsinki and the CONSORT 2010 Statement. All study participants provided written informed consent to participate in the trial.

We enrolled consecutive patients from August 2017 till October 2020. A considerable delay in recruitment was caused by a relative rare occurrence of calcaneal fractures suitable for intramedullary nailing and due to the COVID-19 pandemic. Adults aged more than 18 years undergoing intramedullary nail implantation for a calcaneal fracture were eligible for participation. Exclusion criteria were as follows: a history of allergy to local anesthetics and analgesics used in the study, any contraindication to the sciatic nerve block such as infection at the puncture site, chronic opioid use, any contraindication to the intramedullary nailing technique, use of tramadol before the procedure, and inability to obtain or lack of consent for inclusion in the study.

Patients were randomly assigned in a 1:1 ratio to receive either 100 mg tramadol intramuscularly (tramadol group) or normal saline intramuscularly (control group). The random allocation sequence was generated by a single research assistant (Grzegorz Gorniewski) before the start of participant recruitment with sequentially numbered, opaque, and sealed envelopes. Prior to sciatic nerve block, identical syringes were prepared by the single research assistant (Grzegorz Gorniewski) not directly involved in patient care in a manner blinded to other participants of the study. All participants, attending anesthesiologists and outcome assessors were not informed of the group allocation.

### 2.1. Anesthetic Procedure and Intervention

No premedication was administered prior to arrival in theatre. Standard monitoring included electrocardiography, peripheral pulse oximetry and noninvasive blood pressure (NBP) measurement. Intravenous midazolam (Midanium, Polfa Warszawa, Warsaw, Poland) of 1 to 3 mg was administered to patients after confirming consent for the procedures and study inclusion. All participants received a spinal anesthesia with hyperbaric bupivacaine 10–15 mg (Marcaine Spinal 0.5% Heavy, Aspen Pharma Trading Limited, Dublin, Ireland). Pre-emptive analgesia with intravenous Paracetamol 1 g (Paracetamol Kabi Deutschland GmbH, Friedberg, Germany) was started at the time of performing the spinal anesthesia. Following onset of the spinal block, a single-shot sciatic nerve block was performed in the popliteal region at the division of the tibial and common peroneal nerve. The patient’s lower limb was flexed at the hip and knee joint to obtain access to the popliteal area and a linear 8–12 MHz transducer was used to identify the point of injection. Under sterile conditions, an 80 mm 22-gauge needle (Stimuplex D, B. Braun Melsungen AG, Melsungen, Germany) tip was positioned in-plane initially under the point of division of the sciatic nerve with the aim to spread the solution inside the perineural membrane of the sciatic nerve. On confirmation of appropriate spread a total of 20 mL of 0.25% bupivacaine was injected. All sciatic nerve blocks were performed by experienced anesthesia providers (Marek Janiak, Rafal Kowalczyk).

Patients randomized to the active treatment group received an intramuscular injection with 100 mg of tramadol (Poltram 100, Polpharma S.A., Starogard Gdanski, Poland) and patients in the control group received a similar volume of normal saline solution intramuscularly. The injections were performed into the non-operated lower limb that was blocked by the spinal anesthesia making it pain-free. The spinal anesthesia was performed in all cases from a sitting position, followed by an immediate supine patient placement and therefore the hemodynamic effects of the spinal block on both limbs were considered equal. Intramuscular tramadol absorption can be assumed to be identical and independent in which lower limb the drug was administered.

Postoperatively, all patients had intravenous paracetamol 1 g continued every 6 h starting from the pre-emptive dose and metamizole (Pyralgin, Polpharma S.A., Starogard Gdański, Poland) 1 g every 6 h beginning at block resolution (Numerical Rating Score, NRS > 0). Morphine hydrochloride was administered at doses of 1 mg kg^−1^ (but not more than 10 mg) every 4 to 6 h beginning with a NRS > 3 and continued as nurse-controlled analgesia (NCA).

### 2.2. Intramedullary Nailing

Before surgery, all patients received a single dose of preoperative antibiotic prophylaxis (1.5 g cefuroxime intravenous). Positioning of the patient for the surgery was supine with the operated limb placed outside the surgical table on a strut. The surgical procedure was performed by the same technique and by the same experienced surgical team (Piotr Wasilewski) using a dedicated calcaneal nail set (Calcanail^®^, FH Orthopedics, Heimsbrunn, France), making the surgery standard for all patients in this study. No wound drains were required, and no additional cast was needed.

### 2.3. Study Endpoints

The primary endpoint was the time to first sensation of pain in the operated limb (NRS > 0). The participants and nursing staff were asked to note the time of first pain. This request was made at several points of the study: at the moment of taking consent for the trial, at arrival to the theatre, and on transfer to the postoperative unit.

Secondary outcomes were time to first request for morphine (NRS > 3), total 24 h morphine consumption, and total time of sensory block. The total analgesic requirements were taken from the patients’ medical charts.

### 2.4. Statistical Analysis

Our sample size calculation was based on an expected block prolongation of at least 50% making it clinically meaningful. This would lead to an extension of a pain-free period of at least 2 to 3 h. Any shorter time to first pain would not be considered clinically relevant. According to our power analysis (alpha = 0.05 and beta = 0.9), a sample size of 18 participants was required.

Statistica 13 (Dell Incorporated, Tulsa, OK, USA) was used for the statistical analysis. All variables were tested for normality of distribution via the Shapiro–Wilk test. Parametric variables were reported as mean ± SD and non-parametric variables as median [IQR] with comparison between groups made by a student *t*-test or a Mann–Whitney U test. Categorical data were reported as number (proportion in percentage) with comparisons evaluated using the chi square test. A *p* value of less than 0.05 was considered statistically significant. Kaplan-Meier plots and the log-rank test were used because of a significant proportion of censored observations in time to first opioid analysis.

## 3. Results

Of 43 patients assessed for eligibility, two did not meet the inclusion criteria due to known tramadol intolerance and one declined to participate, leaving 40 for enrolment. One patient in the treatment group retracted the consent due to objection to the intramuscular injection, and two were excluded due to a protocol breach. Finally, 19 participants in the tramadol group and 18 in the control group were included in the analysis. The flowchart according to the CONSORT statement is summarized in Figure 1. There were no complications of the performed blocks and for all the patients, the postoperative period was overall uneventful.

The mean age of the patients at the time of surgery was 48 ± 13.3 (ranging from 21 to 80). There were 10 women and 27 men, which were comparably divided between the two groups. Baseline personal and clinical characteristics were comparable between the groups (Table 1), with one patient assessed as American Society of Anesthesiologists’ physical status III in the control group. All subjects in the study had a unilateral closed calcaneal fracture, with over 70% having intra-articular involvement. Based on the Sanders classification of calcaneal fractures, the most common type was Sanders A (lateral third two-part fracture), followed by Sanders type B (two-part, central third of calcaneum), which is in line to the results of the study that assessed the epidemiology of 957 of such fractures in CT scans [1]. In our study, no differences in pain assessment were noted between different types of calcaneal fractures, but this would not be statistically significant due to the number of subjects in the study.

Significant variability in the duration of the sensory block was seen, ranging from 310 min to over 24 h independent of the group allocation. For the primary endpoint, the median time to first sensation of pain with NRS > 0 was 670 min in the tramadol group and 578 min in the control group. This difference was not clinically and statistically significant (*p* = 0.17, log-rank test) as seen in the Kaplan-Meier plot (Figure 2). No significant difference in the time to first morphine request for an NRS > 3 using the survival analysis with the log-rank test (*p* = 0.21) was observed although a visible trend in favor of the treatment group was noted (Figure 3).

The cumulative morphine consumption in the 24 h following the surgery was assessed with the mean for the tramadol group 0.066 mg kg^−1^ compared with 0.125 mg kg^−1^ in the control group. This difference was found not to be statistically significant (*p* = 0.057 log-rank test).

## 4. Discussion

In our study, intramuscular administration of tramadol with a single-shot sciatic nerve block did not prolong the duration of the sensory blockade beyond the cutoff time of at least 2 h as measured by the time to first simple analgesia and the first opioid administration, which makes our initial observation of two patients with an extended block an incidental finding. The sciatic nerve block extended the analgesia beyond the spinal block which would suggest that all our study participants had an effective nerve block.

The role of adjuvants in peripheral nerve blocks is to optimize the analgesic properties of the local anesthetic by enhancing the onset of action and extending the duration of the sensory block, thereby decreasing analgesic requirements, especially the cumulative doses of opioids. Several adjuvants, such as dexamethasone and dexmedetomidine, have shown a beneficial effect when administered perineurally, but the exact mechanisms of action are debated. Focus on the possible systemic absorption of such drugs and its effect on other pain pathway points is considered and superiority of perineural over systemic administration is not yet confirmed. Administration of 0.15 mg/kg or 8 mg of intravenous dexamethasone in combination with a peripheral nerve block has provided pain relief for 75% of patients after ankle and shoulder surgery for more than 24 h [6]. A study by Abdallah et al. [7] showed that the alfa 2 agonist, dexmedetomidine, extended the analgesia of an interscalene plexus block to about 10 h compared with the placebo independent of whether administered as a perineural or intravenous adjuvant. Alternative methods of extending the duration of analgesia such as continuous nerve block techniques with the aid of catheters or repeat single-shot nerve blocks are cumbersome and require additional health resources which make the use of pharmacological adjuvants the first choice in present regional anesthesia practice.

Tramadol is not clearly recommended as an adjuvant to nerve blocks as unequivocal evidence for its efficacy is lacking and potential for side effects not negligible with little known on its neurotoxic potential when administered perineurally [5], though it remains a commonly used opioid analgetic in many postoperative settings. A direct local anesthetic effect of tramadol was observed in animal models [8,9] as well as in human volunteers [10], but the evidence for the effect of tramadol on nerve blockade is unclear. It may be important to note that the local anesthetic effect similar to 2% lidocaine was observed when tramadol was administered to a sciatic nerve of a rat and its role as an adjuvant in this setting warranted trial designs similar to ours. No previous trial has been performed evaluating the clinical effect of systemic tramadol as an adjuvant in sciatic nerve blocks in human subjects, making our trial one of the first to transfer the possible results seen in the animal models to a clinical scenario. There exist some evidence of its capability to prolong clinically relevant analgesia when used as a block adjuvant to levobupivacaine [11], lidocaine [12], mepivacaine [13], and ropivacaine [14,15] for different blocks and surgical procedures. Analgesia prolongation with brachial plexus block has been observed with perineural [16] and intramuscular administration [11], but not necessarily with intravenous administration of tramadol [17]. In the study by Alemanno et al., intramuscular tramadol at a dose of 1.5 mg/kg in conjunction with a single-shot interscalene block had a similar effect to the perineural administration and prolonged the time to first analgesia request in patients undergoing an intense pain generating arthroscopic rotator cuff repair [11]. The authors of this trial suggested that some of the conflicting results from other studies may have been due to studies involving tramadol as an adjuvant in surgery generating mild postoperative pain and therefore impairing accuracy in detecting differences between study groups, but in our study the expected postoperative pain levels are high [3]. In a systematic review for tramadol as an adjunct in brachial plexus blocks, the analgesia prolongation effect, although relatively reliable, remained highly heterogenous between the studies and with a median of 125.5 min [16]. Tramadol has been reported to improve analgesia of the periprostatic nerve block during urological procedures [18], inferior alveolar nerve block in endodontic surgery [19].

Sciatic nerve block alone has a relatively long duration of action, with use of 20 mL of levobupivacaine or ropivacaine, depending on concentration its analgesic efficacy effective analgesia lasts for 13 to 19 h with higher concentrations resulting in longer motor and sensory blockade [20]. Similar mean duration was observed for 0.5% bupivacaine of 880 min when it was compared with other local anesthetics such as ropivacaine [21]. The observed median time in the tramadol group of our study of 670 min for the sciatic nerve block was shorter and of considerable variation, which could not be explained by any of the collected variables including the type of the calcaneal fracture. Block duration did not correlate in our study with bupivacaine dose per kilogram of patient body weight.

Opioids are well-studied in preemptive or preventive analgesia although the quality of evidence remains moderate [22]. Tramadol seems to be effective in both settings, with administration of the tramadol before induction of anesthesia resulting in lower total consumption of the analgesic [23,24]. In orthopedic surgery, tramadol with paracetamol seems to be efficient in preemptive or preventive analgesia administered both oral or intravenous [25,26], as well as intraarticular [27] or as subcutaneous wound infiltration [28]. The timing of tramadol administration in our study before the surgical insult was not aimed at assessing the possible preemptive analgesic mechanism, but was based on the standard use of several known and effective nerve block adjuvants such as dexamethasone administered at the time of performing the block itself.

Intramuscular injection is generally not a recommended route for tramadol, but no clear evidence of perineural injection of tramadol over systemic administration has been demonstrated. In our study, patients were not able to feel any pain of injection as spinal anesthesia was performed before the intramuscular injection. The duration of analgesic effect of a single 100 mg tramadol dose is about 6 h after intravenous or oral administration and the duration of action after intramuscular injection seems to be similar with tramadol administered via an intramuscular injection having a bioequivalent availability as a 30 min intravenous infusion with peak systemic concentration at 1.1 h from the time of administration [29]. The duration of action of intramuscular tramadol alone does not fully explain the observed delay to first opioid analgesic request in our study treatment group, although residual analgesic activity of single 100 mg intramuscular tramadol dose may extend over 6 h [30]. A perineural administration of tramadol was considered prior to trial design, but at the time of the study, tramadol use in combination with local anesthetic agents for nerve blocks was off label and would have an unknown, yet possible neurotoxic effect on the sciatic nerve in human trial participants with a potential risk outweighing any analgesic benefit.

There seems to exist some potentiation effect of systemically administered tramadol and local anesthesia similar to the systemic effect of dexamethasone on local and regional anesthesia, especially as a part of multimodal analgesia [31,32]. Tramadol has multiple mechanisms of antinociception including activation of μ opioid receptors, inhibition of neuronal reuptake of norepinephrine and serotonin with several metabolites exerting the mentioned effects [29]. Our study shows the potential for tramadol in the setting of combining with regional anesthesia but does not support its block elongating effect after intramuscular administration.

Strengths of our study are the reduced variability of cofounders due to uniform population of patients, operated by one team with identical technique and blocks performed by selected experienced providers with uniform ultrasound guided technique. The use of varying dosage of hyperbaric bupivacaine of 10 to 15 mg for the spinal anesthesia in our trial was unlikely to affect the results, as the spinal block covers a period of up to 4 h from administration of the drug prior to surgery with non-significant differences in the spinal block times as seen in the trial by Axelsson et al. [33]. The 4 h period is a lot shorter than the sciatic blockade and shorter than the times to first analgesia. The delivered drug remains as a 0.5% solution and therefore 10–15 mg is 2–3 mL administered intrathecally, with the dose adjusted to the height of the patient. Adjustment of dose to height is special in spinal anesthesia as most dosing regimens are based on patient weight.

Study shortcomings are the relatively small size, inability to detect effects smaller than assumed that may have been masked by the long duration of the sciatic nerve block itself, a long acquisition period, and also both the bupivacaine used for the nerve block and tramadol dose were not adjusted to patient body weight. Sensory and motor block onset times of the sciatic nerve were not assessed as the spinal blockade masked any possibility of testing, which could have led to a possible missed sciatic nerve block failure.

## 5. Conclusions

In conclusion, a concomitant intramuscular single dose of 100 mg of tramadol does not extend the duration of the analgesic effect of a sciatic nerve block with 20 mL of 0.25% bupivacaine beyond a clinically relevant postulated 2 h in patients undergoing calcaneal fracture nailing. Our study did not demonstrate statistically relevant longer times to first opioid administration in patients receiving intramuscular tramadol, although such a trend could be seen. An opioid sparing effect could not be demonstrated in our trial. Sciatic nerve blockade for calcaneal fracture nailing may provide in itself a clinically relevant analgesic effect.

## Figures and Tables

**Figure 1 healthcare-11-00498-f001:**
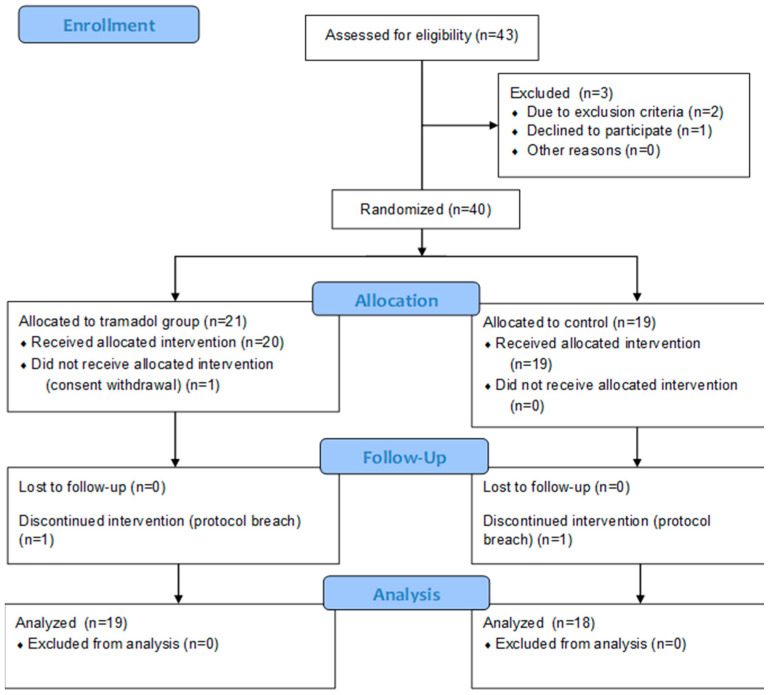
CONSORT flowchart.

**Figure 2 healthcare-11-00498-f002:**
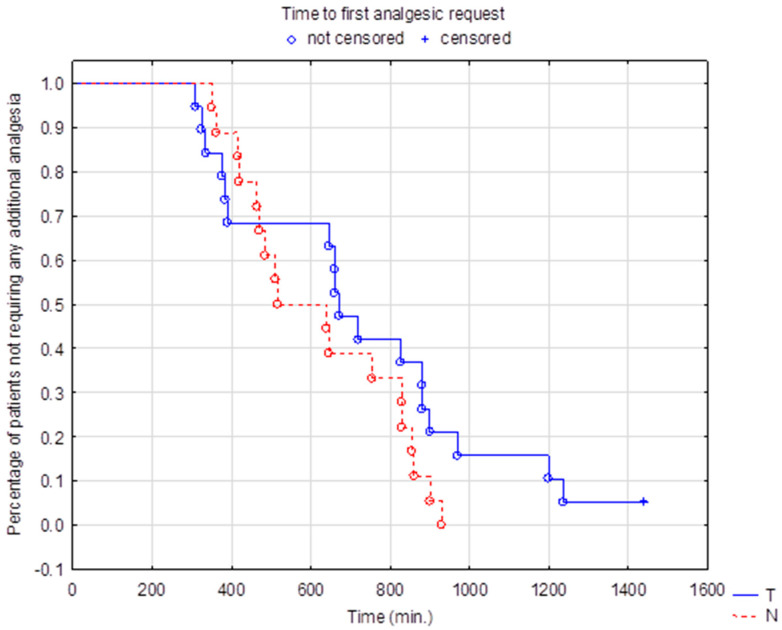
Kaplan-Meier plot for time to first analgesic request for NRS > 0. Blue line (T): tramadol group. Red line (N): control group.

**Figure 3 healthcare-11-00498-f003:**
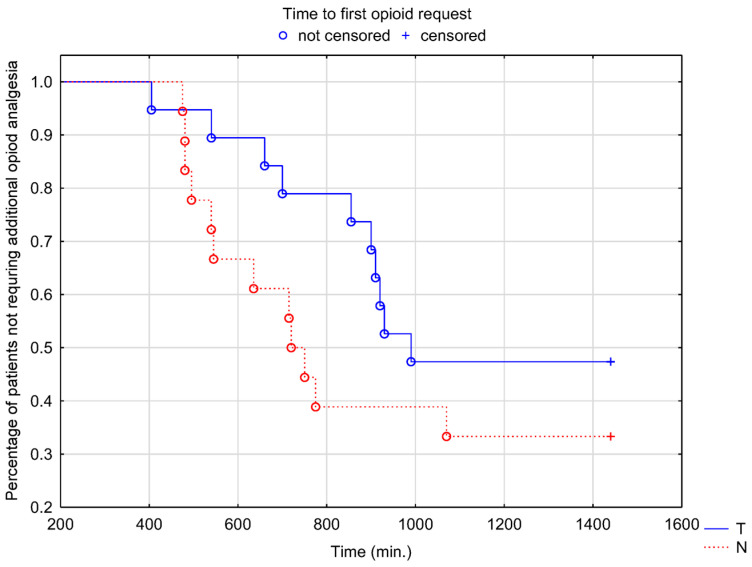
Kaplan–Meier plot of time to first requested opioid for NRS > 3. Blue line (T): tramadol group. Red line (N): control group.

**Table 1 healthcare-11-00498-t001:** Basic demographic characteristics.

Variables	Tramadol Group N = 19	Control Group N = 18	*p* Value
Age (years)	51 ± 10.28 (31–66)	45 ± 15.8 (21–80)	*p* = 0.22 (*t*-test)
Height (cm)	172.2 ± 12.26 (140–189)	171.1 ± 8.11 (169–190)	*p* = 0.74 (*t*-test)
Weight (kg)	77.2 ± 15.22 (57–132)	72.4 ± 12.68 (48–95)	*p* = 0.31 (*t*-test)
ASA physical status (I/II/III)	9/10/0	8/9/1	*p* = 0.99 (Chi2)
Duration of surgery (minutes)	50 [45–55]	58 [50–70]	*p* = 0.15 (Mann–Whitney U)
Sex (M/F)	4/15 (21%/78%)	6/12 (33%/57%)	*p* = 0.4 (Chi2)

Values are reported as mean ± SD, median [IQR] or number (percentage) where appropriate. ASA: American Society of Anesthesiologists.

## Data Availability

Not applicable.

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
