# Peer review of "Effect of Intramuscular Tramadol on the Duration of Clinically Relevant Sciatic Nerve Blockade in Patients Undergoing Calcaneal Fracture Fixation: A Randomized Controlled Trial"

_healthcare, 2023, doi:10.3390/healthcare11040498_

Round 1

Reviewer 1 Report

In this randomized controlled trial, the authors aimed to investigate whether an incidental finding of an extension of a sciatic block with intramuscular administration of 100 mg of tramadol after calcaneal fracture fixation could be confirmed. The tramadol group received a sciatic nerve block with 20 ml of 0.25% bupivacaine and a concomitant dose of 100 mg of intramuscular tramadol, while the control group received an identical sciatic nerve block with concomitant injection of normal saline (placebo). The authors concluded that intramuscular administration of tramadol does not prolong the duration of the sensory blockade as measured by the time to first sensation of pain and the time to the first request for opioid analgesia. Therefore, sciatic nerve blockade may provide in itself a clinically relevant analgesic effect without any need for adjuvants.

My comments are as follows:

1.     Why there was a variation in the dose of hyperbaric bupivacaine used for spinal anesthesia? According tot the authors, 10-15 mg of bupivacaine were used. This variation could have resulted to different durations of blocks in patients, differences in time to first sensation of pain (if some sciatic blocks were not successful- the authors themselves acknowledge the fact that the spinal blockade masked any possibility of testing of the sciatic block) and thus introduce bias…could the authors explain the rationale for this range in bupivacaine dose?

2.     In the statistics section please mention the fact that you used the chi square test for comparison of categorical variables between the two groups. You are mentioning the statistical tests used for assessment of the numerical variables but not for categorical variables.

3.     In line 175, I think it would be preferable to use the phrase: “the median time to first sensation of pain”. This way, you would be in accordance with the primary endpoint mentioned in the line 124

4.     Could you explain the rationale for intramuscular administration of tramadol? You are mentioning the fact that intramuscular administration is not a recommended route for tramadol and in fact, the duration of action after intramuscular administration has been shown to be similar to intravenous administration

Author Response

My comments are as follows:

  1. Why there was a variation in the dose of hyperbaric bupivacaine used for spinal anesthesia? According tot the authors, 10-15 mg of bupivacaine were used. This variation could have resulted to different durations of blocks in patients, differences in time to first sensation of pain (if some sciatic blocks were not successful- the authors themselves acknowledge the fact that the spinal blockade masked any possibility of testing of the sciatic block) and thus introduce bias…could the authors explain the rationale for this range in bupivacaine dose?

Dear Reviewer, it is highly unlikely that the dose of bupivacaine could affect the results of this trial, as spinal anesthesia covers a period of up to 4 hours from the administration of the drug prior to surgery with non significant time differences as seen in the trial of Axelsson et al. [1]. This period is a lot shorter than the sciatic blockade and shorter than the times to first analgesia. The delivered drug remains as a 0.5% solution and therefore 10-15 mg is 2-3 ml administered intrathecally, with the dose adjusted to the height of the patient. Adjustment of dose to height is special in spinal anesthesia, different to most dosing regimens based on patient weight.

[1] Axelsson KH, Edström HH, Sundberg AE, Widman GB. Spinal anaesthesia with hyperbaric 0.5% bupivacaine: effects of volume. Acta Anaesthesiol Scand. 1982 Oct;26(5):439-45. doi: 10.1111/j.1399-6576.1982.tb01796.x. PMID: 6183916.

  1. In the statistics section please mention the fact that you used the chi square test for comparison of categorical variables between the two groups. You are mentioning the statistical tests used for assessment of the numerical variables but not for categorical variables.

This a valuable point and will be corrected in the statistics description.

  1. In line 175, I think it would be preferable to use the phrase: “the median time to first sensation of pain”. This way, you would be in accordance with the primary endpoint mentioned in the line 124

We agree. This will be addressed as is suggested.

  1. Could you explain the rationale for intramuscular administration of tramadol? You are mentioning the fact that intramuscular administration is not a recommended route for tramadol and in fact, the duration of action after intramuscular administration has been shown to be similar to intravenous administration

The use of intramuscular administration was intentional for 2 reasons:

  1. The incidental finding of 2 patients with extensive block had intramuscular tramadol administration.
  2. The intramuscular route was chosen as it resembles perineural administration in pharmacokinetics i.e. absorption, distribution, peak systemic concentrations. The use of tramadol as a perineural adjuvant was considered, but this is off label use and the drug is not registered for such an administration.

We thank You for the suggestions and comments.

Reviewer 2 Report

The premise of the study was based on just two patients. An observational study would have been a better alternative rather than a direct RCT. If an RCT was to be performed, the number of participants could have been more.

the sample size calculation requires more detailing.

Author Response

The premise of the study was based on just two patients. An observational study would have been a better alternative rather than a direct RCT. If an RCT was to be performed, the number of participants could have been more.

We do agree that an observational trial may have been more appropriate in retrospect. However, the use of a randomized controlled trial provides stronger clinical evidence and calcaneal fracture fixation is not a common surgical procedure – the RCT has reduced the time from the question to answer and no participant experienced a complication. The trial was designed to answer one main question of analgesic elongation for at least 24 hours favoring 'overnight' analgesia. Increasing the number of participants would not change the answer to the set hypothesis.

the sample size calculation requires more detailing.

With sciatic nerve block duration lasting 19.3h +/- SD 3 as reported in literature [1] and an expected increase in the mean to at least 24 hours, with the beta set to 0.9, alpha at 0.05 and enrolment ratio of 1 – power calculations provide a desired number of participants at 18.

[1] Sinha, S. A., Mutha, S. C., & Phalgune, D. S. (2016). Efficacy of Sciatic Nerve Block for Pain Management in below Knee Orthopaedic Surgery. Journal of clinical and diagnostic research : JCDR, 10(9), UC17–UC20. https://doi.org/10.7860/JCDR/2016/20418.8496

We thank You for the suggestions and comments.

Reviewer 3 Report

I have read with great interest this paper, that is well structured. However, I have some concerns whether  IM tramadol was administered in the same opertaed  leg or the contralateral, since tha authors state that IM administration resembled perineural one.

Could you please specify?

Moreover, the Im as the authors underline, is not approved for tramadol. So how they justify this way of administration?

In advance, the duration of analgesic effect of tramadol is 6 h, and the duration of the block is almost longer, since the median time for analgesic request is 600 min! Also sensory block lasted 5 h, even if  with high variability.

With this premise, how tramadol could increase  the analgesic effect of the block?

Author Response

I have read with great interest this paper, that is well structured. However, I have some concerns whether  IM tramadol was administered in the same opertaed  leg or the contralateral, since tha authors state that IM administration resembled perineural one.

The intramuscular dose was administered in the contralateral lower limb. The use of hyperbaric bupivacaine for the spinal anesthesia and identical supine positioning of all participants cause a similar hemodynamic effect on both lower limbs. Intramuscular tramadol absorption can therefore be assumed to be identical and independent where the drug was administered.

Moreover, the Im as the authors underline, is not approved for tramadol. So how they justify this way of administration?

We stated that perineural administration is not approved i.e. administering tramadol in combination with the local anesthetic in the fascial plane in the close vicinity of the sciatic nerve. Although, the absorption profile would be the same, the possible unknown but possible neurotoxic effect of tramadol is not without relevance.

In advance, the duration of analgesic effect of tramadol is 6 h, and the duration of the block is almost longer, since the median time for analgesic request is 600 min! Also sensory block lasted 5 h, even if  with high variability.

With this premise, how tramadol could increase  the analgesic effect of the block?

We do agree that the time of analgesic effect should be considered. However, several adjuvants have an opioid sparing and analgesic effect exceeding the expected drug duration. For example: dexamethasone has a half life of 4 hours [1], but its administration as an adjuvant in regional anesthesia allows for a post-orthopedic surgery opioid sparing effect lasting up to 24 hours which is well beyond the expected time of action of a single dose [2].

[1] Johnson DB, Lopez MJ, Kelley B. Dexamethasone. [Updated 2022 May 15]. In: StatPearls [Internet]. Treasure Island (FL): StatPearls Publishing; 2022 Jan-. Available from: https://www.ncbi.nlm.nih.gov/books/NBK482130/?report=classic

[2] Patacsil JA, McAuliffe MS, Feyh LS, Sigmon LL. Local Anesthetic Adjuvants Providing the Longest Duration of Analgesia for Single- Injection Peripheral Nerve Blocks in Orthopedic Surgery: A Literature Review. AANA J. 2016 Apr;84(2):95-103. PMID: 27311150

Thank You for the review.

Reviewer 4 Report

I would like to thank the authors for their work.

I think that the main limit of the study could be a lack of originality, considering that previous trials analyzed the role of tramadol as adjuvant in loco-regional blocks.

I would like to ask to the authors:

- Planning the study, did the authors decide to analyze the role of tramadol as adjuvant of the block or as part of preventive analgesia? Which was the first hypothesis? 

It is not completely clear why they decided to use the intramuscular route as way of administration and how they selected the time of injection of tramadol.

All the other limits (for example a small sample size) are well underlined in the discussion. 

Author Response

I think that the main limit of the study could be a lack of originality, considering that previous trials analyzed the role of tramadol as adjuvant in loco-regional blocks.

No study has reported the use of tramadol as an intramuscular adjuvant in sciatic nerve blocks. Its intramuscular effect has shown brachial plexus block extension by Alemanno et al. [position 9 of manuscript references].

I would like to ask to the authors: 

- Planning the study, did the authors decide to analyze the role of tramadol as adjuvant of the block or as part of preventive analgesia? Which was the first hypothesis?

The main hypothesis was to consider tramadol as an adjuvant in sciatic nerve blocks. The evidence of tramadol is conflicting with several studies showing a prolongation of block duration, but several meta-analysis show no effect in peripheral nerve blocks. Our study may add to this knowledge.

It is not completely clear why they decided to use the intramuscular route as way of administration and how they selected the time of injection of tramadol.

The timing of tramadol administration was based on the standard use of several known and effective adjuvants such as dexamethasone – its use at a dose of 8mg administered at the time of sciatic nerve blockade, as in our trial, has shown to provide even a 24 hour post orthopedic surgery opioid sparing effect (Fredrickson Fanzca MJ, Danesh-Clough TK, White R. Adjuvant dexamethasone for bupivacaine sciatic and ankle blocks: results from 2 randomized placebo-controlled trials. Reg Anesth Pain Med. 2013;38(4):300-307, Patacsil JA, McAuliffe MS, Feyh LS, Sigmon LL. Local Anesthetic Adjuvants Providing the Longest Duration of Analgesia for Single- Injection Peripheral Nerve Blocks in Orthopedic Surgery: A Literature Review. AANA J. 2016 Apr;84(2):95-103. PMID: 27311150.)

 The use of intramuscular administration was intentional for 2 reasons: 1. The incidental finding of 2 patients with extensive block had intramuscular tramadol administration. 2. The intramuscular route was chosen as it resembles perineural administration in pharmacokinetics i.e. absorption, distribution, peak systemic concentrations. The use of tramadol as a perineural adjuvant could have been considered, but this is off label and the drug is not registered for such an administration.

All the other limits (for example a small sample size) are well underlined in the discussion.

Thank You for the review.

Round 2

Reviewer 3 Report

The paper is much improved.

However, I still do not understand in which way intramuscular  injection ( on the other side of the operated one!!) could be comparable to perineurial injection, and therefore why it was chosen for this trial.

Maybe it's better to remove this consideration.

Moreover, the  sciatic block duration could significantly  have masked with any clinical effect of tramadol, so one again I do not understand the pharmacokinetic premises of your study.

Author Response

However, I still do not understand in which way intramuscular  injection ( on the other side of the operated one!!) could be comparable to perineurial injection, and therefore why it was chosen for this trial.

Dear Reviewer, we have removed the above consideration from the manuscript as we agree that this may be controversial as no direct studies exist on the pharmacokinetics of adjuvants administered perineurally. We have added a paragraph in the discussion section highlighting the role of adjuvants in peripheral nerve blockades with systemic effects of dexamethasone and dexmedetomidine. 

Moreover, the  sciatic block duration could significantly  have masked with any clinical effect of tramadol, so one again I do not understand the pharmacokinetic premises of your study.

The exact mechanisms by which adjuvants exert their effect is still debated. As the mechanism by which known and adopted into clinical practice adjuvants such as dexamethasone administered systematically exert a beneficial extension of analgesia is quite often discovered by chance; we do not have a direct explanation of the possible mechanism - but as stated in the manuscript two cases of a 24 hour pain free period, prior to the trial design sparked our interest and led us to conduct this randomized clinical study.

Thank You for the review.